# Atomic-thick metastable phase RhMo nanosheets for hydrogen oxidation catalysis

Juntao Zhang[1,8], Xiaozhi Liu[2,8], Yujin Ji[3,8], Xuerui Liu[4], Dong Su[2]✉,
Zhongbin Zhuang[4], Yu-Chung Chang[5], Chih-Wen Pao[5], Qi Shao[6]✉,
Zhiwei Hu[7] & Xiaoqing Huang[1]✉

Metastable phase two-dimensional catalysts provide great flexibility for modifying their chemical, physical, and electronic properties. However, the synthesis of ultrathin metastable phase two-dimensional metallic nanomaterials is highly challenging, mainly due to the anisotropic nature of metallic materials and their thermodynamically unstable ground-state. Here, we report free-standing RhMo nanosheets with atomic thickness and a unique core/shell (metastable phase/stable phase) structure. The polymorphic interface between the core region and shell region stabilizes and activates metastable phase catalysts; the RhMo Nanosheets/C shows excellent hydrogen oxidation activity and stability. Specifically, the mass activities of RhMo Nanosheets/C is 6.96 A $mg_{Rh}^{-1}$; this is 21.09 times higher than that of commercial Pt/C (0.33 A $mg_{Pt}^{-1}$). Density functional theory calculations suggest that the interface aids in the dissociation of $H_2$ and the H species can then spillover to weak H binding sites for desorption, providing excellent hydrogen oxidation activity for RhMo nanosheets. This work advances the highly controlled synthesis of two-dimensional metastable phase noble metals and provides great directions for the design of high-performance catalysts for fuel cells and beyond.

Two-dimensional (2D) materials, such as graphene[1,2], transition metal dichalcogenides (TMDs)[3,4], black phosphorus[5,6], and layer double hydroxides (LDHs)[7,8], are considered promising material platforms for fabricating the catalysts with superb activity due to their unique electronic structures and intrinsic physical properties compared with those of their bulk materials[9,10]. Over the past years, 2D materials have displayed great promise in many applications such as electronics[11,12], sensors[13], catalysis[14], and beyond. In general, their high maximal surface-to-volume rates[15,16], numerous unsaturated atoms[17,18], and excellent conductivity[2,19], are highly advantageous for catalytic applications, especially for noble-based materials. However, the majority of current 2D materials are limited to van der Waals materials, in which the intralayer interaction is a strong covalent bond, whereas the interlayer interaction is a relatively weak van der Waals bond[14,20]; these van der Waals materials are relatively easy to use to fabricate of 2D structures. Conversely, noble metal atoms prefer to form a 3D close-packed structure owing to the anisotropic nature of metallic materials, causing in extreme difficulty in the synthesis of ultrathin 2D noble materials. Therefore, synthesizing ultrathin 2D noble materials is highly important but challenging.

Along with engineering the 2D structure, regulating the metastable phase is one of the most promising ways to optimize catalytic

[1]State Key Laboratory of Physical Chemistry of Solid Surfaces, College of Chemistry and Chemical Engineering, Xiamen University, Xiamen 361005, China. [2]Beijing National Laboratory for Condensed Matter Physics, Institute of Physics, Chinese Academy of Sciences, Beijing 100190, China. [3]Institute of Functional Nano and Soft Materials (FUNSOM), Jiangsu Key Laboratory for Carbon-Based Functional Materials & Devices, Soochow University, Jiangsu 215123, China. [4]State Key Lab of Organic-Inorganic Composites, Beijing University of Chemical Technology, Beijing 100029, China. [5]National Synchrotron Radiation Research Center, 101 Hsin-Ann Road, 30076 Hsinchu, Taiwan. [6]College of Chemistry and Chemical Engineering and Materials Science, Soochow University, Jiangsu 215123, China. [7]Max Planck Institute for Chemical Physics of Solids, Nothnitzer Strasse 40, Dresden 01187, Germany. [8]These authors contributed equally: Juntao Zhang, Xiaozhi Liu, Yujin Ji. ✉e-mail: dongsu@iphy.ac.cn; qshao@suda.edu.cn; hxq006@xmu.edu.cn

activity since the electronic structure is strongly associated with the crystal configuration[21–24]. Different from the thermodynamically stable state structure, the metastable phase structure has preferable performance in various fields[25,26]. For instance, Zhang and coworkers found that metastable face-centered cubic (fcc) phase Ru nanocrystals exhibited higher hydrogen evolution reaction (HER) performance than hexagonal close-packed (hcp) phase Ru[27]. However, the synthesis of metastable phase 2D noble metal structures remain extremely difficult, since the thermodynamic instability of metastable phase materials and the principles for preparing metastable phase materials are largely heuristic. Therefore, designing metastable phase Rh-based materials while simultaneously achieving ultrathin 2D materials can construct catalysts with high performance and be highly challenging.

Herein, we report a class of uniform free-standing metastable phase RhMo nanosheets (RhMo NSs) with atomic thickness. More importantly, detailed characterizations show a unique core/shell (metastable/stable phase) structure within the core of the metastable hcp phase RhMo alloy and the shell of the stable fcc phase RhMo alloy. Based on the theoretical simulations, the polymorphic interface significantly decreases the formation energy of the whole system, resulting in the enhanced stability of metastable hcp phase RhMo. By integrating atomic thickness and the polymorphic interface, RhMo NSs/C delivers excellent HOR performance in a rotating disk electrode (RDE) and hydroxide exchange membrane fuel cells (HEMFCs). In

particular, the mass and specific activities of RhMo NSs/C in RDE are 6.96 A $mg_{Rh}^{-1}$ and 3.73 A $cm^{-2}$, respectively; these are approximately 36.63 and 10.97 times those of Rh/C (0.19 A $mg_{Rh}^{-1}$ and 0.34 A $cm^{-2}$), and 21.09 and 7.04 times those of commercial Pt/C (0.33 A $mg_{Pt}^{-1}$ and 0.53 A $cm^{-2}$). Moreover, for HEMFC, the RhMo NSs/C delivers high peak power densities (PPDs) of 1.52 W $cm^{-2}$ in $H_2/O_2$ and 0.85 W $cm^{-2}$ in $H_2$/air ($CO_2$-free) conditions, much higher than those of commercial Pt/C (1.25 and 0.48 W $cm^{-2}$, respectively). Additionally, the RhMo NSs/C-based HEMFC only suffers 16% voltage loss after a constant current density of 0.5 A $cm^{-2}$ for 30 h in $H_2$-air ($CO_2$-free) conditions.

## Results

### Morphological and structural characterizations

The ultrathin RhMo NSs were prepared by a facile one-pot wet-chemical method, in which dodecarbonyltetrarhodium ($Rh_4(CO)_{12}$) and molybdenum carbonyl ($Mo(CO)_6$) were used as metal precursors, potassium bromide (KBr) and citric acid monohydrate ($C_7H_8O·H_2O$, CA) as the structure-directing agents, and oleylamine (OAm) as surfactant and solvent (details in the Methods). High-angle annular dark-field scanning transmission electron microscopy (HAADF-STEM) and TEM images showed the uniform hexagonal NSs with the synthetic yield approaching 100% (Fig. 1a and Supplementary Fig. 1). The NSs were uniformly laid flat on the TEM grid (Fig. 1b, c), indicating the ultrathin nature of the NS. According to the statistics from the TEM

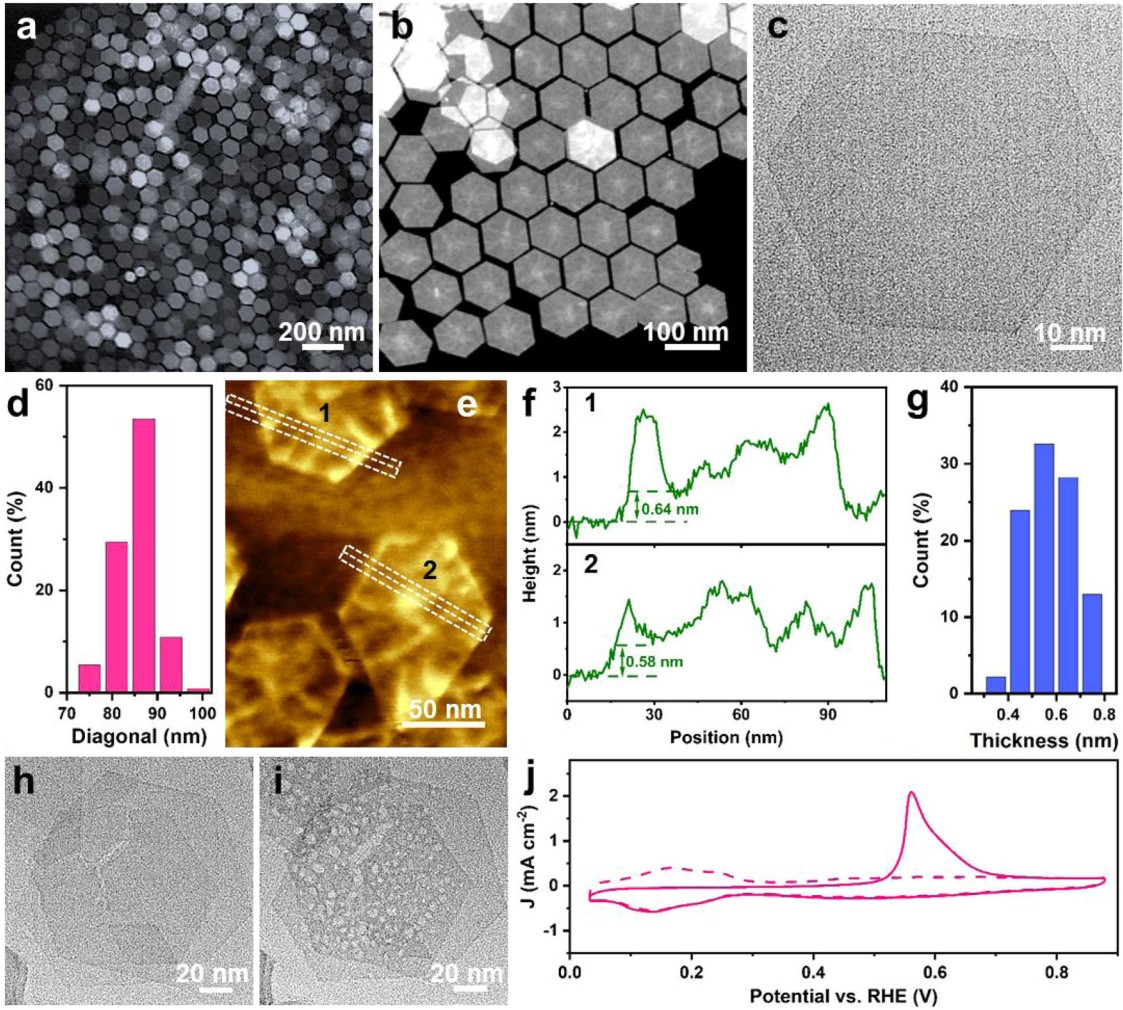

**Fig. 1 | Morphology characterizations of RhMo NSs. a, b** HAADF-STEM images, **c** TEM image, **d** diagonal length distribution, **e** AFM image with **f** the corresponding height profiles and **g** thickness distribution of RhMo NSs. TEM images after

exposing RhMo NSs under electron beam irradiation for **h** 0 s and **i** 5 s. **j** CO stripping voltammograms of RhMo NSs.

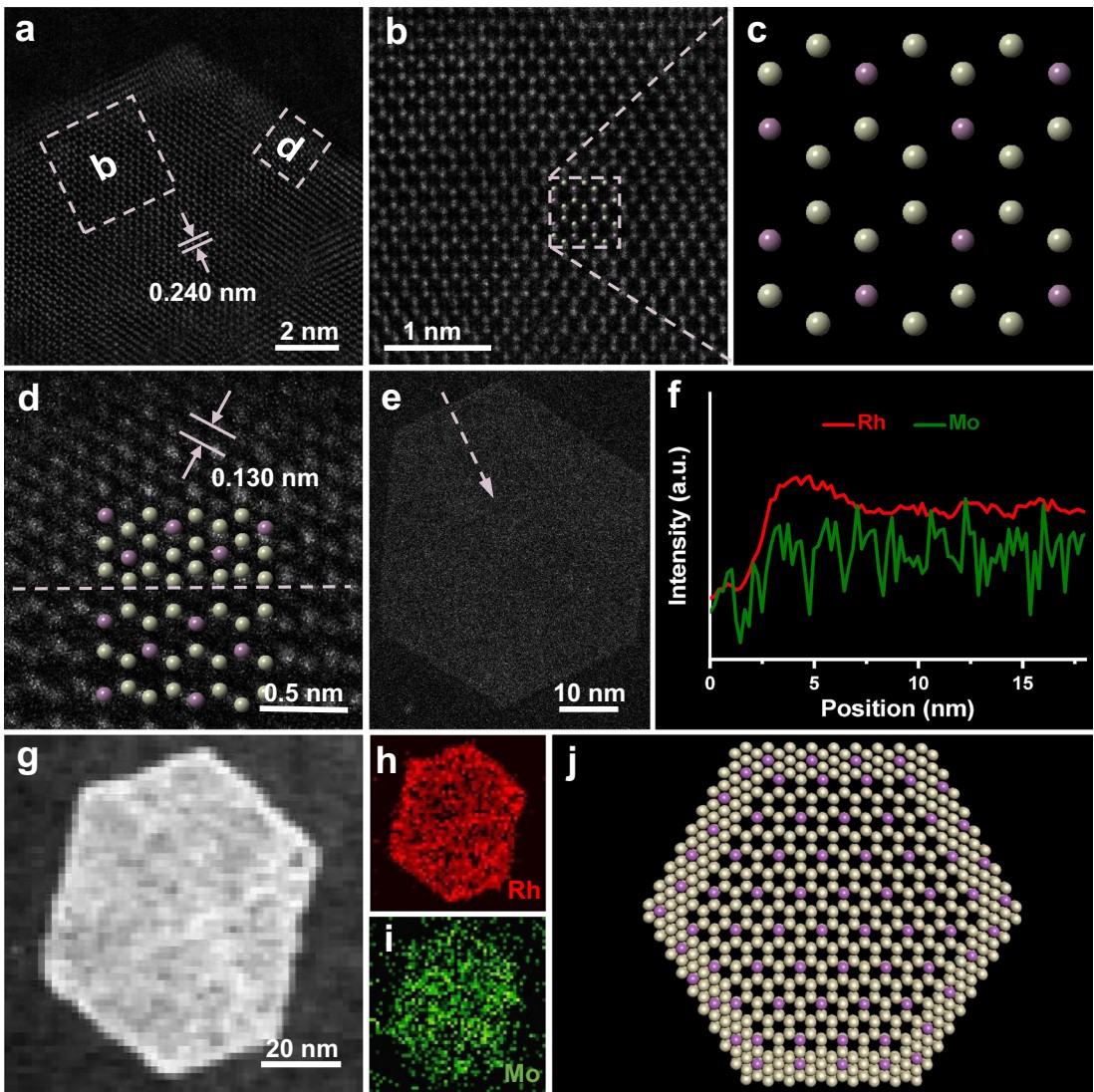

**Fig. 2 | Structural analysis of RhMo NSs. a** HAADF-STEM image from a single RhMo NS. **b** and **d** are high-resolution HAADF images from the remarked areas in (**a**). **c** is a model of atomic arrangement remarked in (**b**). **e** HAADF-STEM image and **f** corresponding EELS spectroscopy line-scan profiles of a single RhMo NS. **g–i** STEM-EELS elemental mapping of RhMo NSs; Rh mapping in red and Mo mapping in green. **j** The schematic atom models of RhMo NSs showing the top view, where white and purple ball represent Rh and Mo atom.

image, the mean diagonal length distribution of RhMo NAs was 85.3 nm (Fig. 1d). The superthin nature was also confirmed by atomic force microscopy (AFM) analysis. As shown in Fig. 1e–g and Supplementary Fig. 2, the thickness of NSs was measured to be 0.57 nm, being consistent with the TEM results (Supplementary Fig. 3). Compared with the TEM image of RhMo NSs, the TEM image of RhMo NSs displayed obvious white cavities after exposure to electron beam for 5 s (Fig. 1h, i), further confirming the ultrathin character of RhMo NSs. Moreover, the superthin structure was shown in the high electrochemical surface area (ECSA). As shown in Fig. 1j, the ECSA of RhMo NSs, determined by means of CO stripping, exhibited the highest ECSA of 185.2 m$^2$ g$_{Rh}^{-1}$, which was much larger than those of Rh/C (53.2 m$^2$ g$_{Rh}^{-1}$) and Pt/C (62.5 m$^2$ g$_{Pt}^{-1}$) (Supplementary Fig. 4). The X-ray diffraction pattern (XRD) of RhMo NSs displayed hardly diffraction peaks, further indicating the superthin structure (Supplementary Fig. 5). STEM-energy dispersive X-ray spectroscopy (STEM-EDX) (Supplementary Fig. 6) revealed that the ratio of Rh/Mo was 84:16, in accordance with the inductively coupled plasma atomic emission spectroscopy (ICP-AES) result (85:15). The oxidation state of Rh in RhMo NSs was studied by X-ray absorption near-edge spectra (XANES)

at the Rh-K edge, which was highly sensitive to the valence state of the 4d element since the absorption edge shifted by more than one eV when the valence state increased by one[28,29]. The energy position of RhMo NSs spectrum shifts by more than 4 eV to lower energy relative to Rh$^{3+}$ reference (Rh$_2$O$_3$), but was located very close to that of Rh foil, indicating a basic metallic nature of Rh in RhMo NSs (Supplementary Fig. 7); these results were consistent with the X-ray photoelectron spectroscopy (XPS) results (Supplementary Fig. 8)

To further elucidate the crystal structure of RhMo NSs, aberration-corrected HAADF-STEM (AC-HAADF-STEM) was carried out. Figure 2a displays the AC-HAADF-STEM image of RhMo NSs; the lattice spacing of 0.240 nm in core regions (slightly larger than that of hcp Rh (0.236 nm)) can be assigned to the [1 0 -1 0] facet of the hcp RhMo alloy. The atoms in the core region are arranged in a six-membered ring structure, as the unique structural feature of the hexagonal phase, and can be indexed to a typical [1 0 -1 0] zone axis of P63/mmc (Fig. 2c, magnified in Fig. 2b). Additionally, the selected-area electron diffraction (SAED) of a single NS displays a single crystalline crystal of the hexagonal phase (Supplementary Fig. 9), which is consistent with the AC-HAADF-STEM results. These results confirm the hcp nature of core regions.

Notably, a polymorphic interface is clearly observed in Fig. 2d (marked in Fig. 2a); its atomic arrangement of edge regions is different from that of the core regions. In detail, the lattice fringe with interplanar spacing in the shell region is 0.130 nm (slightly smaller than that of Rh (0.135 nm)), which is assigned to the [2 2 0] facet of fcc RhMo (PDF#87-0714), implying an the fcc structure of Rh in the shell region. Furthermore, the element distributions in NSs were investigated by STEM-electron energy-loss spectroscopy (STEM-EELS) line-scan and mapping analysis. As shown in Fig. 2e, f, the content of Rh in the edge region is slightly higher than that in the core region, since the density of atoms in the shell region is higher than that in the core region (directly observed in Fig. 2d). Additionally, a distinct core/shell structure was observed in the AC-HAADF-STEM image (Fig. 2g), showing the existence of a polymorphic interface. The corresponding mapping images of Rh (red) and Mo (green) show that the content of Rh in shell region is slightly higher than that in core region, while Mo is dispersed throughout the whole NS (Fig. 2h–i). From these results, we can conclude that NSs have characteristics of an atomic thickness and a unique polymorphic interface between the metastable hcp phase RhMo alloy in the core region and the stable fcc RhMo alloy in the shell region (Fig. 2j).

To further investigate the formation mechanism of RhMo NSs, we analyzed the morphologies and compositions of reaction intermediates collected after different time intervals by using TEM, and SEM–EDS (Supplementary Fig. 10). At the early synthetic stage (10 min), the 2D nanocrystals with the average edge length of 25 nm is obtained. From 10 min to 120 min, the small nanosheets gradually grow along the in-plane direction (Supplementary Fig. 10a–e). The ratios of Rh/Mo were almost the same during the reaction process (Supplementary Fig. 10f). Supplementary Fig. 10g displays the formation process of RhMo NSs, which involves the initial formation of small free-standing NS, with the subsequent growth along the in-plane direction to the final product.

Wet-chemical approaches provide an effective way for preparing core–shell nanostructures[30,31]. However, considering the unique metastable phase of RhMo NSs, a set of control experiments were explored to understand how to tune the nanostructures. When replacing $Mo(CO)_6$ with $MoCl_6$ while keeping other synthetic parameters the same, only irregular nanoparticles (NPs) were observed (Supplementary Fig. 11). Simultaneously, NPs associated with irregular NSs were produced when reducing the dosage of $Mo(CO)_6$, suggesting the significance of enough CO for the growth of well-defined NSs (Supplementary Fig. 12). Importantly, this method can also be extended to the synthesis of other free-standing Rh-based NSs by replacing $Mo(CO)_6$ with other carbonyl compounds ($W(CO)_6$, $Cr(CO)_6$, and $Fe(CO)_5$, Supplementary Fig. 13). It is suggested that the strong adsorption of CO molecules on the basal planes of nanosheets prevents growth along the basal direction and is responsible for directing the formation of the sheet-like structure[32]. Therefore, the growth of free-standing RhMo NS highly depends on the released CO in the synthetic process. Simultaneously, the use of KBr is also critical for growing the well-defined NSs. As shown in Supplementary Fig. 14a, b, only irregular NSs were obtained without using KBr. Meanwhile, with insufficient amount of KBr, low-quality rhombic NSs were obtained (Supplementary Fig. 14c, d). The excessive KBr decrease the size of NSs (Supplementary Fig. 14e, f). Moreover, the low-quality free-standing NSs can also be produced by replacing KBr with KCl and cetyltrimethyl ammonium bromide (Supplementary Fig. 15). This is attributed to halide ions (e.g., $Br^-$ and $Cl^-$) can be used to control the growth of well-defined NSs[32]. Besides, the concentrations of CA also had to stay within a critical range to fabricate a high-quality RhMo NSs (Supplementary Fig. 16).

## HOR performance
The electrochemical properties of RhMo NSs were studied to determine the HOR in $H_2$-saturated 0.1 M KOH by using a typical three-electrode system. For comparison, Rh/C and commercial Pt/C (Johnson Matthey, 20 wt% Pt, Supplementary Fig. 17) were selected as the references. Prior to the electrochemical test, the RhMo NSs were loaded on commercial carbon powder (Vulcan XC-72R) via sonication of the RhMo NSs and C (RhMo NSs/C, Supplementary Fig. 18). Supplementary Fig. 19 shows the cyclic voltammograms (CVs) of these samples in $N_2$-saturated 0.1 M KOH electrolyte at a scan rate of 50 mV s$^{-1}$; the peak of the underpotentially deposited hydrogen ($H_{upd}$) of RhMo NSs/C was much larger than those of Rh/C and commercial Pt/C, suggesting the maximum utilization of Rh in RhMo NSs/C. Additionally, the $H_{upd}$ peak of RhMo NSs/C was negatively shifted compared to that of Rh/C and commercial Pt/C, showing a weaker adsorption of Rh−H binding on RhMo NSs/C, which may be beneficial for the HOR process[33,34].

The HOR polarization curves in Fig. 3a reveal that RhMo NSs/C exhibited the highest HOR performance among these catalysts. Specifically, the current density of RhMo NSs/C at an overpotential of 50 mV is 2.66 mA cm$^{-2}$, which is much higher than those of Pt/C (1.96 mA cm$^{-2}$), and Rh/C (1.47 mA cm$^{-2}$). Considering that no anodic current is observed in $N_2$-saturated 0.1 M KOH electrolyte above 0 V, we conclude that the anodic current density in Fig. 3a is mainly derived from $H_2$ oxidation (Supplementary Fig. 20). Moreover, the polarization curves of RhMo NSs/C at different rotating speeds are obtained (Fig. 3b). As calculated with the Koutechy−Levich method shown in Supplementary Fig. 21, the calculated slope of RhMo NSs/C is 13.12 cm$^2$ mA$^{-1}$ rpm$^{1/2}$, suggesting that the anode current is mainly derived from the two-electron transfer reaction. Figure 3c displays the Tafel plots of the kinetic current ($i_k$) on RhMo NSs/C, Rh/C and commercial Pt/C; the RhMo NSs/C displays the highest $i_k$ among these catalysts at various potentials.

To further compare with other catalysts, we normalized the mass activity by mass loading and the specific activity by electrochemical surface area (ECSA). RhMo NSs/C shows a mass activity of 6.96 A mg$_{Rh}^{-1}$ at a potential of 50 mV versus RHE, which is 36.63 and 21.09 times higher than those of Rh/C (0.19 A mg$_{Rh}^{-1}$) and commercial Pt/C (0.33 A mg$_{Pt}^{-1}$) (Fig. 3d), respectively; this mass activity is much higher than those of previously reported Rh-based and Pt-based HOR electrocatalysts (Fig. 3e and Supplementary Table 1). Although having the highest ECSA, RhMo NSs/C has a specific activity of 3.73 A cm$^{-2}$, which is 10.97 and 7.04 times higher than those of Rh/C (0.34 A cm$^{-2}$) and Pt/C (0.53 A cm$^{-2}$), respectively, revealing the significance of the metastable phase of Rh-based materials in promoting the HOR process. The RhMo NSs/C also displayed the highest exchange current of HOR/HER by fitting with the Butler-Volmer equation (Supplementary Fig. 22).

The HOR durability of RhMo NSs/C, Rh/C and commercial Pt/C was evaluated by performing a chronoamperometry test on RDE at an overpotential of 100 mV with a continuous rotation speed of 1600 rpm. As shown in Supplementary Fig. 23, RhMo NSs/C can retain approximately 87% of its original current density after 20,000 s continuous test. In contrast, Rh/C and Pt/C only maintain about 29 and 21% of their original current densities, respectively. Additionally, the NS structure of spent RhMo NSs/C was maintained whereas Rh/C and Pt/C were severely agglomerated (Supplementary Fig. 24). The aforementioned results show that RhMo NSs/C are active and stable towards the alkaline HOR. The RhMo NSs/C can also display promising hydrogen evolution reaction (HER) performance, in which it needs a low overpotential of 34 mV to reach a current density of −10 mA cm$^{-2}$ in alkaline media (Supplementary Fig. 25).

## HEMFCs performance
Considering the superior HOR performance of RhMo NSs/C at the RDE level, the practical $H_2$-$O_2$ and $H_2$-air ($CO_2$-free) HEMFC performances and durability were also conducted by the fuel cell device. RhMo NSs/C (0.2 mg$_{Rh}$ cm$^{-2}$), commercial Pt/C (0.2 mg$_{Pt}$ cm$^{-2}$), and PAP-TP-85 were employed as the anode, cathode and membrane for fabricating the

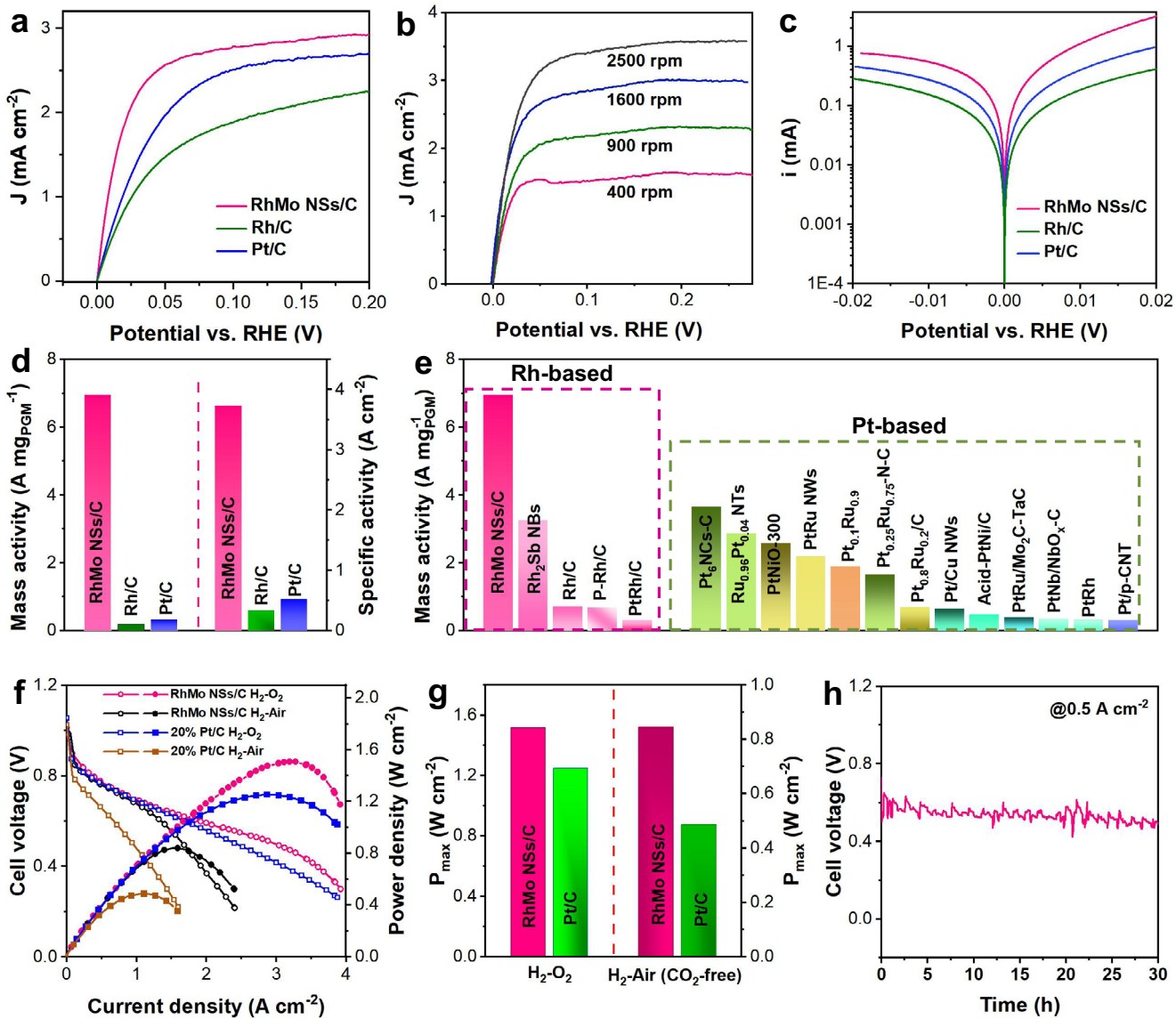

**Fig. 3 | HOR and MEA evaluations of RhMo NSs. a** HOR polarization curves of RhMo NSs/C, Rh/C, and Pt/C in 0.1 M KOH with a scan rate of 5 mV s$^{-1}$ at a rotation speed of 1600 rpm. **b** HOR polarization curves of RhMo NSs/C with different rotation speeds. **c** Tafel slopes of RhMo NSs/C, Rh/C, and Pt/C. **d** Mass and specific activities of RhMo NSs/C, Rh/C, and Pt/C. **e** Comparison of the mass activity of RhMo NSs/C and previously reported electrocatalysts at an overpotential of 50 mV in 0.1 M KOH. **f** Polarization curves and peak power density curves of HEMFCs with RhMo NSs NSs/C (0.2 mg$_{Rh}$ cm$^{-2}$) or commercial Pt/C (0.2 mg$_{Pt}$ cm$^{-2}$) in anode and commercial Pt/C (0.2 mg$_{Pt}$ cm$^{-2}$) in cathode under H$_2$/O$_2$ and H$_2$/air (CO$_2$-free). **g** The PPD$_{max}$ of RhMo NSs/C-based and commercial Pt/C-based MEA under H$_2$−O$_2$ and H$_2$/air (CO$_2$-free) media. **h** H$_2$-air/(CO$_2$-free) HEMFC stability test at the current density of 500 mA cm$^{-2}$ with RhMo NSs/C (0.2 mg$_{Rh}$ cm$^{-2}$) in anode and commercial Pt/C (0.2 mg$_{Pt}$ cm$^{-2}$, 60 wt% Pt/C) in cathode.

membrane electrode assembly (MEA). The reference MEA was fabricated by using commercial Pt/C as the anode (0.2 mg$_{Pt}$ cm$^{-2}$) and cathode (0.2 mg$_{Pt}$ cm$^{-2}$). Figure 3f displays the polarization curves and peak power density (PPD) curves of HEMFCs for RhMo NSs NSs/C and commercial Pt/C under H$_2$/O$_2$ and H$_2$/air (CO$_2$-free). The MEA catalyzed by RhMo NSs/C clearly delivers a higher current density and power density. In detail, the RhMo NSs/C-based MEA delivers high current densities of 1357 and 1132 mA cm$^{-2}$ at 0.65 V (typical operating potential for automotive applications) under H$_2$/O$_2$ and H$_2$/air (CO$_2$-free) conditions, respectively; these current densities are higher than those of commercial Pt/C (1260 mA cm$^{-2}$ and 521 mA cm$^{-2}$). Furthermore, in the H$_2$/O$_2$ gas feed, the MEA with RhMo NSs/C also delivers a PPD of 1.52 W cm$^{-2}$ at a current density of 3.2 A cm$^{-2}$, which is higher than that of commercial Pt/C (1.25 W cm$^{-2}$ at a current density of 2.9 A cm$^{-2}$) (Fig. 3g). A similar situation was observed in the H$_2$/air (CO$_2$-free) conditions. As shown in Fig. 3g, the RhMo NSs/C-based MEA displays a

PPD of 0.85 W cm$^{-2}$, which is higher than that of commercial Pt/C (0.49 W cm$^{-2}$). The MEA fabricated with RhMo NSs/C outperformed previously reported electrocatalysts (Supplementary Table 2), indicating the great potential application of RhMo NSs/C in HEMFCs. The durability of the RhMo NSs/C-based MEA was also investigated. As shown in Fig. 3h, compared with the initial cell voltage, the MEA with RhMo MSs/C only suffered 16% voltage loss after a constant current density of 500 mA cm$^{-2}$ for 30 h, indicating the promising stability of the RhMo NSs/C-based HEMFC.

**DFT calculations**
To shed light on the enhanced HOR performance, density functional theory (DFT) calculations were carried out. We constructed pure hcp, pure fcc and polymorphic RhMo NSs to compare their formation energies per atom (Fig. 4a). The hcp RhMo NS has the highest formation energy (0.9698 eV atom$^{-1}$) due to its metastable nature; the local

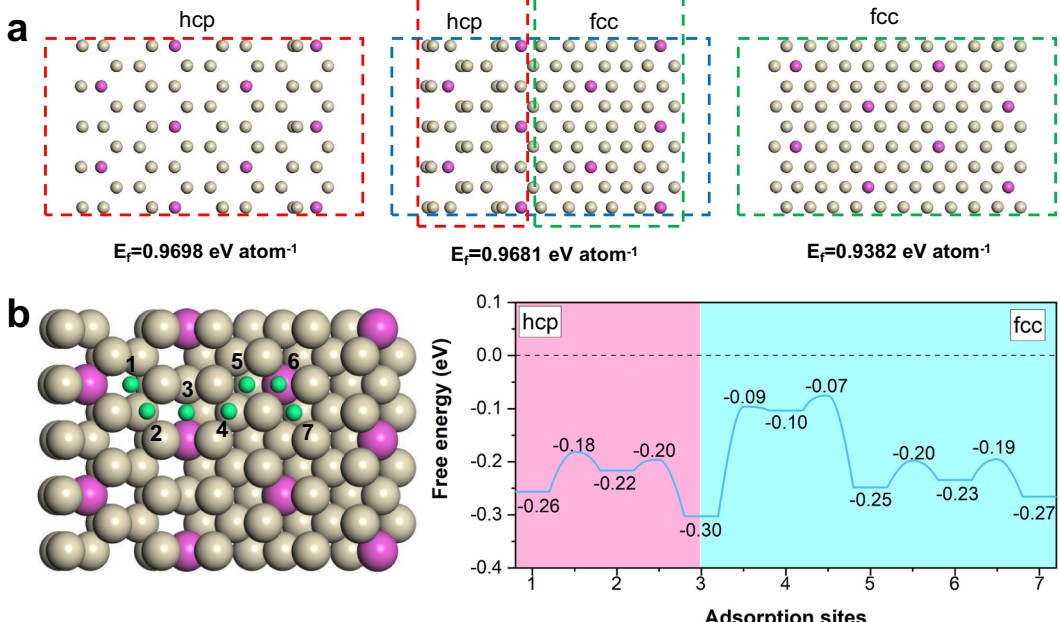

**Fig. 4 | DFT calculations of RhMo NSs. a** The geometric structures for pure hcp, polymorphic, and pure fcc RhMo and their formation energies per atom. **b** Geometry configurations for seven potential active sites on the surface of RhMo NSs and corresponding hydrogen adsorption free energies on each site. Rh, Mo, and H are shown in white, purple, and green balls, respectively.

edge phase transition from metastable hcp phase to stable fcc phase in the RhMo NS can lower the formation energy (0.9681 eV atom⁻¹) to stabilize the whole configuration and therefore protect the inner metastable phase. Furthermore, for HOR electrocatalysts, previous reports indicated that the Volmer step is the rate-determining step regardless of the latter Tafel or Heyrovsky pathway[35,36] and a strong hydrogen binding free energy (HBE) is responsible for the poor HOR activity. Accordingly, as shown in Fig. 4b and Supplementary Fig. 26, we evaluated the HBEs of 7 potential active sites on the surface of RhMo NSs and found that the polymorphic interface between the hcp and fcc phases had a stronger HBE of −0.30 eV, which aided in the dissociation of H₂ to generate *H. In contrast, the hcp and fcc counterparts had weaker HBEs of −0.1 to −0.27 eV, respectively, which facilitated the subsequent desorption of *H into protons. From the hydrogen-rich interface to the hydrogen-poor pure phase RhMo NSs, there may exist a hydrogen spillover pathway to bridge fast *H generation and desorption. Our results showed that the H diffusion on the surface needed to overcome the maximum energy barrier of 0.21 eV (site 3→site 4) for the fcc part and 0.10 eV (site 3→site 2) for the hcp part, indicating the feasibility of hydrogen spillover on the HOR for RhMo NSs. Additionally, the physical mixture of RhMo NSs and WO₃ particles turned dark blue in the H₂ treatment at room temperature, as an evidence of polymorphic interface character (Supplementary Fig. 27). Further electronic analysis reveals the fcc-hcp interface is an electron-rich region with a lower *d* band center, which is helpful for hydrogen adsorption and dissociation while higher *d* band centers on pure fcc phase and hcp phase correspond to weak H binding and facilitate H desorption (Supplementary Fig. 28). Overall, the polymorphic interface in RhMo NSs was the direct reactive site for the HOR and the subsequent hydrogen migration contributed to its high performance.

## Discussion

In summary, we report a class of unique RhMo NSs with atomic thickness and unique core/shell structure; the core is a metastable hcp phase RhMo alloy and the shell is a stable fcc phase RhMo alloy. The atomic thickness enables RhMo NSs/C to have a remarkable high surface area of 185.2 m² g_Rh⁻¹, resulting in a high density of surface-

active sites. In addition, the polymorphic interface between metastable hcp phase RhMo and stable fcc phase RhMo not only stabilizes the metastable hcp structure but also regulates the adsorption state of *H; this interface provides enhanced HOR stability and activity of the RhMo NSs. Consequently, RhMo NSs/C has mass and specific activities of 6.96 A mg⁻¹ and 3.73 A cm⁻² in RDE level. Additionally, HEMFCs with RhMo NSs/C delivers high PPDs of 1.52 W cm⁻² in H₂/O₂ and 0.85 W cm⁻² in H₂/air (CO₂-free) conditions. This work not only provides effective strategy for preparing an ultrathin 2D Rh-based nanomaterials but also emphasizes the significance of polymorphic interface in stabilizing metastable phase noble metals and enhancing catalysis.

## Methods
### Chemicals

Tetrarhodium dodecacarbonyl (Rh₄(CO)₁₂, 98%) and molybdenum carbonyl (Mo(CO)₆, 98%) were purchased from STREM. Rhodium acetylacetonate (Rh(Ac)₃, 97%), rhodium chloride hydrate (RhCl₃·xH₂O, 99%), and molybdenum chloride (MoCl₆, 99%) were purchased from J&K Scientific Ltd. Tungsten hexacarbonyl (W(CO)₆, 99%) was purchased from Macklin Ltd. Hexacarbonylchromium (Cr(CO)₆, 99%) and pentacarbonyl iron (Fe(CO)₅, 97%) were purchased from Alfa Aesar Ltd. Nafion perfluorinated resin solution (5 wt%) was purchased from Sigma Aldrich (USA). Potassium bromide (KBr, AR), citric acid monohydrate (CA, C₆H₈O₇·H₂O, AR), ethanol (C₂H₆O, AR), hexamethylene (C₆H₁₂, AR), isopropanol (C₃H₈O, AR), and potassium hydroxide (KOH, AR) were purchased from Sinopharm Chemical Reagent Co., Ltd. Commercial Pt/C (20 wt% Pt) was purchased from Johnson Matthey (JM) Corporation. All the chemicals were used without further purification. The water (18 MΩ cm⁻¹) used in all the experiments was obtained by passing through an ultra-pure purification system (Aqua Solutions).

### Synthesis of RhMo NSs

For this synthesis, 10 mg Rh₄(CO)₁₂, 50 mg Mo(CO)₆, 32 mg KBr, 100 mg CA and 5 mL oleylamine were mixed into a 30 mL glass vial. After ultrasonication for 1 h, the mixture was heated from room temperature to 160 °C and maintained at 160 °C for 5 h in an oil bath. The

products were collected by centrifugation and washed with a cyclo-hexane/ethanol (1/8) solution several times.

## Synthesis of Rh/C

For the synthesis of Rh/C[37], 20 mg RhCl$_3$ and 50 mg carbon powder were dissolved in deionized water and ethanol and sonicated for 1 h. The mixture was heated at 50 °C to remove the ethanol and then dried by using a freeze dryer. Finally, the powder was annealed at 600 °C with a heating rate of 2 °C/min in an Ar atmosphere.

## Characterization

Transmission electron microscopy (TEM) was performed on a JEOL electron microscope at an accelerating voltage of 100 kV. High-angle annular dark-field scanning TEM (HAADF-STEM), HAADF-STEM energy dispersive X-ray spectroscopy (HAADF-STEM-EDS), and high-resolution TEM (HRTEM) were performed on an FEI Tecnai F30 transmission electron microscope at an accelerating voltage of 300 kV. Scanning electron microscopy EDS (SEM-EDS) was conducted on a ZEISS Sigma scanning electron microscope at an accelerating voltage of 20 kV. Inductively coupled plasma optical spectroscopy (ICP-OES) was carried out on a Thermo Fisher iCAP700 series (Thermo Fisher Scientific, Waltham, MA USA) in axial mode. X-ray diffraction spectroscopy (XRD) was conducted on a Rigaku XRD machine with Cu Kα (λ = 1.540598 Å). X-ray photoelectron spectroscopy (XPS) was conducted on an SSI S-Probe XPS spectrometer. The XANES and EXAFS spectra were measured in transmission mode by using the TPS 44A beamline at the National Synchrotron Radiation Research Centre (NSRRC, Hsinchu).

## Electrochemical measurements

All the electrochemical measurements were performed on the CHI760 electrochemical station (Chenhua, Shanghai) in a typical three-electrode system. Prior to the electrochemical tests, RhMo NSs were dispersed in a 10 mL mixture of hexamethylene and Vulcan XC-72 carbon by sonicating for 1 h to obtain RhMo NSs/C. The final catalysts were obtained by centrifugation and dried at 60 °C. For the preparation of the working electrode, 2 mg RhMo NSs/C, 990 μL isopropanol and 10 μL Nafion solution (5 wt%) were added into a glass vial and then ultrasonicated for 1 h to form a homogenous ink. Then, 7.5 μL ink was transferred onto the glass carbon rotating disk electrode (RDE; Pine Research Instrumentation; diameter, 5 mm; area, 0.196 cm$^2$), resulting in a Rh loading of 0.0176 mg cm$^{-2}$ on GC. The rotation speeds were controlled by the installation of rotating electrode speed control (Pine Research Instrumentation, model: AFMSRCE). A saturated calomel electrode (SCE) and graphite rod were used as the counter electrode and reference electrode, respectively. All the measurements were conducted in 0.1 M KOH solution and the HOR/HER polarization curves were collected at a scan rate of 5 mV s$^{-1}$. For HOR measurements, the mass activities were obtained by calculating the kinetic current at an overpotential of 50 mV (vs. RHE). The stability test was performed in H$_2$-saturated 0.1 M KOH electrolyte though chronoamperometric measurements at an overpotential of 100 mV for 20,000 s. For the CO stripping measurement, the working electrode was immersed in the 0.1 M KOH, in which CO gas (99.99%) was constantly bubbled for 20 min. After purging the above electrolyte with N$_2$ for 20 min, two CV curves were measured at a scan rate of 10 mV s$^{-1}$. The equilibrium potential was the zero point of the HER/HOR by using Pt/C as the working electrode rotating at 1600 rpm in a H$_2$-saturated electrolyte. All the polarization curves were corrected by solution resistance, which was tested by AC-impedance spectroscopy from 200 kHz to 0.1 kHz.

## MEA test

The catalyst ink was fabricated by ultrasonically dispersing the RhMo NSs/C or commercial Pt/C and PAP-TP-100 into deionized water and isopropanol (v/v = 1/20) in an ultrasonic bath for 3 h. The catalyst-coated membrane (CCM) with an electrode area of 5 cm$^{-2}$ was fabricated by spraying the catalyst ink onto both sides of the PAP-TP-85 membrane. Then, the CCM was soaked in a 3 M KOH solution for 12 h and rinsed with deionized water several times to remove the residual KOH. The full HEMFC was assembled with the rinsed CCM, a GDL (SGL 29 BC), a graphite bipolar plate (5 cm$^{-2}$ flow field) and metal current collectors for each side. For the HEMFC performance test, the anode catalyst loading (RhMo NSs/C, 23%) was 0.2 mg$_{Rh}$ cm$^{-2}$ and the cathodic catalyst (commercial Pt/C HiSpec 9100, 40 wt%) loading was 0.2 mg$_{Pt}$ cm$^{-2}$. For the HEMFC durability test, the anodic catalyst loading (RhMo NSs/C, 23%) was 0.2 mg$_{Rh}$ cm$^{-2}$ and the cathodic catalyst loading (commercial Pt/C HiSpec 9100, 60 wt%) loading was 0.2 mg$_{Pt}$ cm$^{-2}$.

## DFT calculations

All theoretical calculations were conducted under the framework of density functional theory (DFT) in the Vienna ab initio software package version 5.4.1;[38] the exchange-correlation energy was calculated using the revised Perdew–Burke–Ernzerhof propose formula[39] due to its better accuracy in chemical adsorption. A projector-augmented wave basis set with an electronic cut-off energy of 400 eV was used to describe the electron-ion interaction. A gamma-centered Monkhorst-Pack grid was sampled for the Brillouin zone integrations with 1 × 2 × 1 k-mesh. During geometry optimization, the thresholds of energy and force corresponded to 1E−4 eV and −0.05 eV/Å, respectively. Based on the computational hydrogen electrode potential model[40], the adsorption free energy of *H was calculated follows:

$$\Delta G(*H) = E(*H) - E(*) - 1/2E(H_2) + 0.29 \tag{1}$$

where $E(*H)$ and $E(*)$ represent the total energies of RhMo NSs with and without H adsorption and $E(H_2)$ is the total energy of the free hydrogen molecule. The effect of the zero-point energy and entropy is considered as constant (0.29 eV for *H). The climbing image nudged elastic band method[41] was used to determine the minimum energy pathway for H diffusion.

## Data availability

The data generated in this study are provided in the Supplementary Information/Source Data file. Source data are provided with this paper.

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

## Acknowledgements

The authors thank the financial supports by the National Key R&D Program of China (2020YFB1505802), the Ministry of Science and Technology of China (2017YFA0208200), the major project of Basic Science (natural science) of Jiangsu Province (21KJA430001), Jiangsu Provincial Natural Science Foundation (BK20211316), the Suzhou Municipal Science and Technology Bureau (SYG202125), State Key Laboratory of Physical Chemistry of Solid Surfaces, Xiamen University (202113), the National Natural Science Foundation of China (22025108, U21A20327, 22121001), China Postdoctoral Science Foundation (2020M682083), the Priority Academic Program Development of Jiangsu Higher Education Institutions (PAPD), Collaborative Innovation Center of Suzhou Nano Science & Technology, and start-up support from Xiamen University.

## Author contributions

X.H. conceived and supervised the research. X.H. and J.Z. designed the experiments and performed most of the experiments and data analysis. Xiaozhi L. and D.S. performed AC-HAADF-STEM test. Y.J. and Q.S. performed the DFT calculation. Xuerui L. and Z.Z. performed the MEA test. Y.C., C.P. and Z.H. performed the XAS experiment. J.Z., Q.S. and X.H wrote the manuscript with support from all co-authors.

## Competing interests

The authors declare no competing interests.
