## [Peer Review File · Nature Communications]

REVIEWER COMMENTS

Reviewer #1 (Remarks to the Author):

This manuscript demonstrates free-standing RhMo nanosheets with only two atomic thickness and unique core/shell structure. The core region in RhMo nanosheet is metastable hcp structure. The author claimed that the polymorphism interface between the metastable hcp phase and the stable fcc phase can not only stabilize the metastable hcp structure but also regulate the adsorption state of *H. The RhMo NSs displays excellent hydrogen oxidation reaction activity and stability in rotating disk electrode and hydroxide exchange membrane fuel cells. The ultrathin RhMo NSs is rarely reported. In general, this work is performed well, and the structure of RhMo NSs is well characterized, and the data can support the conclusion. Therefore, I recommend publishing this work in this journal after the authors addressing some concerns:

1. In this work, the most novelty and fascinating of this manuscript is the highly controlled synthesis of two-dimensional metastable noble metals. It is necessary to help the readers to understand synthesis of 2D metallic materials.
2. The abbreviation of X-ray diffraction pattern is missing in the manuscript.
3. The surface might be oxidized because of the ultrathin nature of RhMo NSs. The author should compare RhMo NSs with RhMo NPs (with fcc phase) to check their surface properties.
4. According to the Figure S22, the Rh/C and Pt/C seems to maintain about 30% and 20% of their initial current density after 20000 s continuous test, not for 78% and 35%.
5. There are some grammatical mistakes need to be corrected. For example:
 - A) On page 3 line 65, "detail characterizations reveal that it shows an unique core/shell (metastable/stable) structure" should be "detail characterizations reveal that it shows a unique core/shell (metastable/stable) structure".
 - B) On page 5 line 102-103, the spellings of "enegry" and "neergy" are not correct, which should be corrected as "energy".

Reviewer #2 (Remarks to the Author):

This is an interesting new work on RhMo nanosheets by the authors. The reported synthesis and resultant interfacial engineering (i.e., hpc/fcc) resulted in enhanced HOR and hydroxide exchange membrane fuel cells (HEMFCs).

The work is of high quality, with all conclusions proven with detailed experiments and DFT calculations. I recommend that it is accepted after the following minor corrections, viz:

1.The first sentence of the abstract “Metastable two-dimensional (2D) structures have provided great flexibility for integrating chemical, physical, and electronic properties of the catalysts” should be re-written to make sense, the grammar is confusing.

2.Also in the abstract, “The polymorphism interface between...” should read “The polymorphic interface between....”.

3.Page 3: “structure, the metastable structure displays the kinetically favored in various fields”. This sentence does not make any grammatic sense, please re-frame the sentence.

4.Page 5: “(XANE)” should read “XANES”

5.Page 5: “the abreaction-corrected...” should be spelt as “aberration-corrected”.

Reviewer #3 (Remarks to the Author):

The authors reported the synthesis and characterization of the 2D hcp-fcc core/shell nanosheet of RhMo alloy. The composite was proved to show much better HOR catalytical performance than commercial Pt/C, Rh/C , as well as many previously reported catalysts (listed in Table S1). While the results are interesting and important, the manuscript was not well organized to show enough salience that makes it acceptable for publication. The reviewer's main concerns and comments are listed as follows for reference.

- 1) The strategy of the synthesis is not new. The authors should cite and review the preceding works, like , e.g. ACS Catal. 2018, 8, 5581–5590; Chem. Rev. 2016, 116, 18, 10414–10472.
- 2) The authors focused on characterizing the atomic-level structure of the material using state-of-the-art techniques. However, the readers may also interest in the mechanism of growth and tuning of the nano structures. Unfortunately, such kind of information was not provided in detail.
- 3) The author calculated and compared the formation energy of the core/shell structure against those of hcp and fcc phases. However, the employed theoretical scheme did not provide deep insight into the electronic structure of the core/shell interface. The review suggest to implement further analysis on the local electronic structures at different regions of the core/shell structure.
- 4) Since the computed supercells for hcp, fcc and hcp/fcc structures are different in sizes, the precision of the PBC theory was not guaranteed to be at the same level for these systems with the same k-mesh. Therefore, directly comparing the formation free energy is dangerous. Moreover, it is rational to compare the formation free energy per atom among the systems, instead of the reported formation energy per supercell.
- 5) The authors calculated the migration barriers of a hydrogen atom at the hcp/fcc interface using NEB method. It is strange that the site5 to site6 step size is much greater than the others. The author should be better to provide in SI the details of the critical structures along the migration path, to ensure that all migration steps are elementary steps along the pathway.
- 6) Too many grammatic errors and typos in the manuscript. Many sentences need to be refined to conform to the logic.

Response to Reviewers

Dear Reviewers,

Thanks for your valuable time to constructive comments on our manuscript titled “Atomic-Thick Metastable RhMo Nanosheets for Hydrogen Oxidation Catalysis” for Nature Communications” (NCOMMS-22-47904-T). We sincerely appreciate your comments and suggestions on our work, which have certainly improved our manuscript. On the basis of all the comments, we have made detailed responses and substantive revisions in the revised manuscript.

Reviewer #1 (Remarks to the Author):

General comment: This manuscript demonstrates free-standing RhMo nanosheets with only two atomic thickness and unique core/shell structure. The core region in RhMo nanosheet is metastable hcp structure. The author claimed that the polymorphism interface between the metastable hcp phase and the stable fcc phase can not only stabilize the metastable hcp structure but also regulate the adsorption state of *H. The RhMo NSs displays excellent hydrogen oxidation reaction activity and stability in rotating disk electrode and hydroxide exchange membrane fuel cells. The ultrathin RhMo NSs is rarely reported. In general, this work is performed well, and the structure of RhMo NSs is well characterized, and the data can support the conclusion. Therefore, I recommend publishing this work in this journal after the authors addressing some concerns:

Response: We greatly appreciate your positive comments on our work, which significantly improve our manuscript to meet the high standard of *Nature Communications*. We have carefully revised our manuscript based on your comments.

Comment 1. In this work, the most novelty and fascinating of this manuscript is the highly controlled synthesis of two-dimensional metastable noble metals. It is necessary to help the readers to understand synthesis of 2D metallic materials.

Response: Thank you very much for your useful comment. According to your suggestion, we explore the growth procedure of RhMo NSs (**Supplementary Fig. 10**). At the early synthetic stage (10 min), the 2D nanocrystals with the average edge length of 25 nm. From 10 min to 120 min, the small nanosheets gradually grow along the in-plane direction (**Supplementary Fig. 10a-e**). The ratios of Rh/Mo were almost the same during the reaction (**Supplementary Fig. 10f**). **Supplementary Fig. 10g** displays the formation process of RhMo NSs, which involves the initial formation of small free-standing NS, with the subsequent growth along the in-plane direction to the final product.

[Added results]: [Page 6, Line 132] “To further investigate the formation mechanism of RhMo NSs, we analyzed the morphologies and compositions of reaction intermediates collected after different time intervals by using TEM, and SEM–EDS (**Supplementary Fig. 10**). At the early synthetic stage (10 min), the 2D nanocrystals with the average edge length of 25 nm is obtained. From 10 min to 120 min, the small nanosheets gradually grow along the in-plane direction (**Supplementary Fig. 10a-e**). The ratios of Rh/Mo were almost the same during the reaction process (**Supplementary Fig. 10f**). **Supplementary Fig. 10g** displays the formation process of RhMo NSs, which involves the initial formation of small free-standing NS, with the subsequent growth along the in-plane direction to the final product.”

Supplementary Fig. 10. Growth mechanism of RhMo NSs. TEM images of intermediates obtained at (a) 10 min, (b) 30 min, (c) 60 min, and (d) 120 min. (e) Edge length and (f) SEM-EDS analysis of the intermediates. (g) Synthetic scheme for RhMo NSs.

Comment 2. The abbreviation of X-ray diffraction pattern is missing in the manuscript.

Response: Thank you very much for your careful review. The abbreviation of XRD diffraction pattern was added into the manuscript.

[Added results]: [Page 5, Line 99] “The X-ray diffraction pattern (XRD)”

Comment 3. The surface might be oxidized because of the ultrathin nature of RhMo NSs. The author should compare RhMo NSs with RhMo NPs (with fcc phase) to check their surface properties.

Response: Thank you very much for your useful comment. According to your suggestion, we compare the XPS results between RhMo NSs and fcc RhMo NPs. As shown in **Supplementary Fig. 8**, the main characteristic peaks of Rh^0 can be observed in RhMo NSs, suggesting the metallic state of Rh in RhMo NSs, be consistent with the X-ray absorption near-edge spectra (XANES) results. In addition, the peak position of Rh^0 in the XPS spectra of RhMo NSs positively shifts to higher binding energy, suggesting that the surface valence state of Rh in RhMo NSs is higher than that of fcc RhMo NPs. Meanwhile, the ratio of $\text{Rh}^{3+}/\text{Rh}^0$ of RhMo NSs is 0.31, which is higher than that of fcc RhMo NPs (0.28), revealing that the ultrathin structure is more tend to oxidized. All the results indicate the surface of ultrathin structure is more easily oxidized than that of nanoparticles.

Supplementary Fig. 8 Rh 3d XPS spectra of fcc RhMo NPs and RhMo NSs.

[Added results]: [Page 5, Line 107] “these results were consistent with the X-ray photoelectron spectroscopy (XPS) results (Supplementary Fig. 8)”

Supplementary Fig. 8 Rh 3d XPS spectra of fcc RhMo NPs and RhMo NSs. The main characteristic peaks of Rh⁰ can be observed in RhMo NSs, suggesting the metallic state of Rh in RhMo NSs, be consistent with the XANES results. In addition, the peak position of Rh⁰ in the XPS spectra of RhMo NSs positively shifts to higher binding energy, suggesting that the surface valence state of Rh in RhMo NSs is higher than that of fcc RhMo NPs. Meanwhile, the ratio of Rh³⁺/Rh⁰ of RhMo NSs is 0.31, which is higher than that of fcc RhMo NPs (0.28), indicating that the ultrathin structure is easy to be oxidized.

Comment 4. According to the Figure S22, the Rh/C and Pt/C seems to maintain about 30% and 20% of their initial current density after 20000 s continuous test, not for 78% and 35%.

Response: Thank you very much for your careful comment. It is correct that after 20000 s continuous test, Rh/C and Pt/C maintained about 29% and 21% of their initial current densities. We have corrected the related results in the manuscript.

[Added results]: [Page 9, Line 195] “In contrast, Rh/C and Pt/C only maintain about 29 and 21% of their original current densities, respectively.”

Comment 5. There are some grammatical mistakes need to be corrected. For example:

A) On page 3 line 65, “detail characterizations reveal that it shows an unique core/shell (metastable/stable) structure” should be “detailed characterizations reveal that it shows a unique core/shell (metastable/stable) structure”.

B) On page 5 line 102-103, the spellings of "enegry" and "neergy" are not correct, which should be corrected as "energy".

Response: Thank you very much for your careful comment, which significantly improve the quality of our manuscript. We have already corrected these mistakes in this manuscript, which was marked in yellow.

[Added results]: [Page 4, Line 67] “detailed characterizations show a unique core/shell (metastable/stable phase) structure”

[Page 5, Line 105] “The energy position of RhMo NSs spectrum shifts by more than 4 eV to lower energy relative to Rh³⁺ reference (Rh₂O₃)”

Reviewer #2 (Remarks to the Author):

General comment: This is an interesting new work on RhMo nanosheets by the authors. The reported synthesis and resultant interfacial engineering (i.e., hcp/fcc) resulted in enhanced HOR and hydroxide exchange membrane fuel cells (HEMFCs). The work is of high quality, with all conclusions proven with detailed experiments and DFT calculations. I recommend that it is accepted after the following minor corrections, viz:

Response: We strongly appreciate your great efforts on our work, which significantly improve our manuscript to meet the high standard of *Nature communications*. We have carefully revised our manuscript based on your comments, and we believe that all those comments have been well addressed in the revised manuscript.

Comment 1. The first sentence of the abstract “Metastable two-dimensional (2D) structures have provided great flexibility for integrating chemical, physical, and electronic properties of the catalysts” should be re-written to make sense, the grammar is confusing.

Response: Thank you very much for your comment. We have modified this sentence to “Metastable phase two-dimensional (2D) catalysts provide great flexibility for modifying their chemical, physical, and electronic properties.”

[Added results]: [Page 2, Line 21] “Metastable phase two-dimensional (2D) catalysts provide great flexibility for modifying their chemical, physical, and electronic properties.”

Comment 2. Also in the abstract, “The polymorphism interface between...” should read “The polymorphic interface between...”.

Response: Thank you very much for your comment. We have corrected in the revised manuscript accordingly.

[Added results]: [Page 2, Line 26] “The polymorphic interface between....”

Comment 3. Page 3: “structure, the metastable structure displays the kinetically favored in various fields”. This sentence does not make any grammatic sense, please re-frame the sentence.

Response: We have modified this sentence to “Different from the thermodynamically stable state structure, the metastable phase structure has preferable performance in various fields”.

[Added results]: [Page 3, Line 57] “Different from the thermodynamically stable state structure, the metastable phase structure has preferable performance in various fields.”

Comment 4. Page 5: “(XANE)” should read “XANES”

Response: Thank you for your careful review. We have corrected in the revised manuscript (**Page 5, Line 12**).

[Added results]: [Page 5, Line 103] “X-ray absorption near-edge spectra (XANES)”

Comment 5. Page 5: “the abreaction-corrected...” should be spelt as “aberration-corrected”.

Response: Thank you for your careful review. We have corrected in the revised manuscript accordingly.

[Added results]: [Page 5, Line 109] “aberration-corrected HAADF-STEM”

Reviewer #3 (Remarks to the Author):

General comment: The authors reported the synthesis and characterization of the 2D hcp-fcc core/shell nanosheet of RhMo alloy. The composite was proved to show much better HOR catalytical performance than commercial Pt/C, Rh/C, as well as many previously reported catalysts (listed in Table S1). While the results are interesting and important, the manuscript was not well organized to show enough salience that makes it acceptable for publication. The reviewer's main concerns and comments are listed as follows for reference.

Response: We greatly appreciate your insightful comments on our work, which significantly improve the quality of our manuscript. First of all, we want to claim that the novelty of our work is to syntheses metastable phase atomic-thick noble metal structures. The preparation of metastable phase atomic-thick noble metal nanomaterials is extremely difficult, since the thermodynamic instability of metastable phase materials and the principles for preparing metastable phase materials are largely heuristic. In this work, we combined the effect from CO, halide ions and reducing agent to successfully obtain the metastable phase atomic-thick RhMo nanosheets.

In addition, we are highly appreciated for your important suggestions. Following your suggestions, we have largely improved the quality of paper based on two aspects:

1. Synthesis study

Following your suggestions, the result about the time tracking experiment is provided (**Page 6, Line 132; Supplementary Fig. 10**). The discussion about the role of different reaction precursors in synthesizing RhMo NSs is also added (**Page 7, Line 141**).

2. Mechanism study

Following your suggestions, we have added the electronic structures and formation free energy per atom of hcp, hcp-fcc and fcc structures (**Figure 4a and Supplementary Fig. 28**). More details about the hydrogen migration path and DFT structures are also provided (**Page 10, Line 225; Page 11, Line 231; Page 11, Line 242; Supplementary Fig. 26 and Supplementary Files**).

Comment 1. The strategy of the synthesis is not new. The authors should cite and review the preceding works, like, e.g. ACS Catal. 2018, 8, 5581–5590; Chem. Rev. 2016, 116, 18, 10414–10472.

Response: Thank you very much for your suggestions. These two important references (ACS Catal. 2018, 8, 5581; Chem. Rev. 2016, 116, 18, 10414.) reported and summarized wet-chemical approaches to prepare core-shell nanostructures. Following your suggestions, we have cited these two works in reference 30 and 31.

Nevertheless, it should be pointed out that both the two mentioned references do not report the preparations of metastable phase nanomaterials, while the novelty of our work is to syntheses metastable phase atomic-thick noble metal structures. The preparation of metastable phase atomic-thick noble metal nanomaterials is extremely difficult, since the thermodynamic instability of metastable phase materials and the principles for preparing metastable phase materials are largely heuristic. In this work, we combined the effect from CO, halide ions and reducing agent to successfully obtain the atomic-thick metastable phase RhMo nanosheets.

[Added results]: [Page 7, Line 140] “Wet-chemical approaches provide an effective way for preparing core-shell nanostructures.^{30,31}”

“30. Qin, Y. *et al.* Intermetallic hcp-PtBi/fcc-Pt core/shell nanoplates enable efficient bifunctional oxygen reduction and methanol oxidation electrocatalysis. *ACS Catal.* **8**, 5581-5590 (2018).

31. Gilroy, K. D., Ruditskiy, A., Peng, H.-C., Qin, D. & Xia, Y. Bimetallic nanocrystals: Syntheses, properties, and applications. *Chem. Rev.* **116**, 10414-10472 (2016).”

Comment 2. The authors focused on characterizing the atomic-level structure of the material using state-of-the-art techniques. However, the readers may also interest in the mechanism of growth and tuning of the nano structures. Unfortunately, such kind of information was not provided in detail.

Response: Thank you very much for your useful comment. According to your suggestion, we have now first explored the growth process of RhMo NSs, shown in **Supplementary Fig. 10**. At the early synthetic stage (10 min), the 2D nanocrystals with the average edge length of 25 nm is obtained. From 10 min to 120 min, the small nanosheets gradually grow along the in-plane direction (**Supplementary Fig. 10a-e**). The ratios of Rh/Mo were almost the same during the reaction process (**Supplementary Fig. 10f**). **Supplementary Fig. 10g** displays the formation process of RhMo NSs, which involves the initial formation of small free-standing NS, with the subsequent growth along the in-plane direction to the final product.

Meanwhile, a set of control experiments were also carried out to tune the nanostructures. When replacing $\text{Mo}(\text{CO})_6$ with MoCl_6 while keeping other synthetic parameters the same, only irregular nanoparticles (NPs) were observed (**Supplementary Fig. 11**). Simultaneously, NPs associated with irregular NSs were produced when reducing the dosage of $\text{Mo}(\text{CO})_6$, suggesting the significance of enough CO for the growth of well-defined NSs (**Supplementary Fig. 12**). Importantly, this method can also be extended to the synthesis of other free-standing Rh-based NSs by replacing $\text{Mo}(\text{CO})_6$ with other carbonyl compounds ($\text{W}(\text{CO})_6$, $\text{Cr}(\text{CO})_6$, and $\text{Fe}(\text{CO})_5$, **Supplementary Fig. 13**). It is suggested that the strong adsorption of CO molecules on the basal planes of nanosheets prevents growth along the basal direction and is responsible for directing the formation of the sheet-like structure (Nat. Nanotechnol. 2011, 6, 28-32). Therefore, the growth of free-standing RhMo NS highly depends on the released CO in the synthetic process.

Simultaneously, the use of KBr is also critical for growing the well-defined NSs. As shown in **Supplementary Fig. 14a, b**, only irregular NSs were obtained without using KBr. Meanwhile, with insufficient amount of KBr, low-quality rhombic NSs were obtained (**Supplementary Fig. 14c, d**). The excessive KBr decrease the size of NSs (**Supplementary Fig. 14e, f**). Moreover, the low-quality free-standing NSs can also be produced by replacing KBr with KCl and cetyltrimethyl ammonium bromide (**Supplementary Fig. 15**). This is attributed to halide ions (e.g., Br^- and Cl^-) can be used to control the growth of well-defined NSs (Nat. Nanotechnol. 2011, 6, 28-32). Besides, the concentrations of CA also had to stay within a critical range to fabricate a high-quality RhMo NSs (**Supplementary Fig. 16**).

Supplementary Fig. 10. Growth mechanism of RhMo NSs. TEM images of intermediates obtained at (a) 10 min, (b) 30 min, (c) 60 min, and (d) 120 min. (e) Edge length and (f) SEM-EDS analysis of the intermediates. (g) Synthetic scheme for RhMo NSs.

Supplementary Fig. 11 (a, b) TEM images, (c) XRD pattern, and (d) SEM-EDS analysis of the products synthesized by replacing $\text{Mo}(\text{CO})_6$ with MoCl_6 .

Supplementary Fig. 12 TEM images of the products synthesized with the same reaction conditions as those of RhMo NSs except the use of (a, b) 0 mg, (c, f) 10 mg, and (e, f) 100 mg $\text{Mo}(\text{CO})_6$.

Supplementary Fig. 13 TEM images of the products with the same reaction conditions as those of RhMo NSs except the use of (a, b) 50 mg $\text{Cr}(\text{CO})_6$, (c, d) 50 mg $\text{W}(\text{CO})_6$, and (e, f) 50 mg $\text{Fe}_2(\text{CO})_9$.

Supplementary Fig. 14 TEM images of products under the typical condition but varying the amounts of (a, b) 0 mg, (c, d) 16 mg, and (e, f) 64 mg KBr.

Supplementary Fig. 15 TEM images of the products synthesized with the same reaction conditions as those of RhMo NSs except replacing KBr with (a, b) CTAB, and (c, d) KCl.

Supplementary Fig. 16 TEM images of the products synthesized with the same reaction conditions as those of RhMo NSs except the use of (a, b) 0 mg, (c, d) 50 mg, and (e, f) 200 mg CA.

[Added results]: [Page 6, Line 132] “To further investigate the formation mechanism of RhMo NSs, we analyzed the morphologies and compositions of reaction intermediates collected after different time intervals by using TEM, and SEM–EDS (**Supplementary Fig. 10**). At the early synthetic stage (10 min), the 2D nanocrystals with the average edge length of 25 nm is obtained. From 10 min to 120 min, the small nanosheets gradually grow along the in-plane direction (**Supplementary Fig. 10a-e**). The ratios of Rh/Mo were almost the same during the reaction process (**Supplementary Fig. 10f**). **Supplementary Fig. 10g** displays the formation process of RhMo NSs, which involves the initial formation of small free-standing NS, with the subsequent growth along the in-plane direction to the final product.”

[Page 7, Line 140] “However, considering the unique metastable phase of RhMo NSs, a set of control experiments were explored to understand how to tune the nanostructures. When replacing $\text{Mo}(\text{CO})_6$ with MoCl_6 while keeping other synthetic parameters the same, only irregular nanoparticles (NPs) were observed (**Supplementary Fig. 11**). Simultaneously, NPs associated with irregular NSs were produced when reducing the dosage of $\text{Mo}(\text{CO})_6$, suggesting the significance of

enough CO for the growth of well-defined NSs (**Supplementary Fig. 12**). Importantly, this method can also be extended to the synthesis of other free-standing Rh-based NSs by replacing Mo(CO)₆ with other carbonyl compounds (W(CO)₆, Cr(CO)₆, and Fe(CO)₅, **Supplementary Fig. 13**). It is suggested that the strong adsorption of CO molecules on the basal planes of nanosheets prevents growth along the basal direction and is responsible for directing the formation of the sheet-like structure.³² Therefore, the growth of free-standing RhMo NS highly depends on the released CO in the synthetic process. Simultaneously, the use of KBr is also critical for growing the well-defined NSs. As shown in **Supplementary Fig. 14a, b**, only irregular NSs were obtained without using KBr. Meanwhile, with insufficient amount of KBr, low-quality rhombic NSs were obtained (**Supplementary Fig. 14c, d**). The excessive KBr decrease the size of NSs (**Supplementary Fig. 14e, f**). Moreover, the low-quality free-standing NSs can also be produced by replacing KBr with KCl and cetyltrimethyl ammonium bromide (**Supplementary Fig. 15**). This is attributed to halide ions (e.g., Br⁻ and Cl⁻) can be used to control the growth of well-defined NSs.³² Besides, the concentrations of CA also had to stay within a critical range to fabricate a high-quality RhMo NSs (**Supplementary Fig. 16**).”

Supplementary Fig. 10. Growth mechanism of RhMo NSs. TEM images of intermediates obtained at (a) 10 min, (b) 30 min, (c) 60 min, and (d) 120 min. (e) Edge length and (f) SEM-EDS analysis of the intermediates. (g) Synthetic scheme for RhMo NSs.

Comment 3. The author calculated and compared the formation energy of the core/shell structure against those of hcp and fcc phases. However, the employed theoretical scheme did not provide deep insight into the electronic structure of the core/shell interface. The review suggests to implement further analysis on the local electronic structures at different regions of the core/shell structure.

Response: Thank you very much for your useful comment. According to your suggestion, we plotted the charge densities of difference polymorphism RhMo structures to show its local electron properties. As shown in **Supplementary Fig. 28a**, there exists an electron-rich region at the interface of fcc-hcp phase, which is helpful for the hydrogen adsorption and dissociation. Then we analyzed the d orbital distribution of the sites A, D and J (**Supplementary Fig. 26a**) to represent the hcp, hcp-fcc and fcc phases and found that the hcp-fcc interface has a lower band center, which corresponds to the stronger adsorption than those of the pure hcp and fcc phases (**Supplementary Fig. 28b-d**). Thus, the hcp-fcc interface on RhMo NSs is responsible for hydrogen dissociation while the pure fcc and hcp phases are more helpful for hydrogen evolution. Accordingly, we provided this explanation in manuscript.

Supplementary Fig. 28 (a) Charge density difference plot of polymorphism RhMo structures and the projected d bands on site A, D and J to represent the electronic properties of (b) hcp (site A), (c) hcp-fcc (site D) and (d) fcc regions (site J).

[Added results]: [Page 11, Line 242] “Further electronic analysis reveals the fcc-hcp interface is an electron-rich region with a lower d band center, which is helpful for hydrogen adsorption and dissociation while higher d band centers on pure *fcc* phase and *hcp* phase correspond to weak H binding and facilitate H desorption (**Supplementary Fig. 28**).”

Supplementary Fig. 28 (a) Charge density difference plot of polymorphism RhMo structures and the projected d bands on site A, D and J to represent the electronic properties of (b) hcp (site A), (c) hcp-fcc (site D) and (d) fcc regions (site J).

Comment 4. Since the computed supercells for hcp, fcc and hcp/fcc structures are different in sizes, the precision of the PBC theory was not guaranteed to be at the same level for these systems with the same k-mesh. Therefore, directly comparing the formation free energy is dangerous. Moreover, it is rational to compare the formation free energy per atom among the systems, instead of the reported formation energy per supercell.

Response: Thank you very much for your useful comment. Following your suggestions, we reported the formation energy per atom (E_f) in **Fig. 4a**. It is found that the E_f s of hcp, hcp-fcc and fcc structures are 0.9698, 0.9681 and 0.9382 eV atom⁻¹, respectively. Thus, it also supports our conclusion that the partial phase transition is helpful for the stabilization of hcp-fcc RhMo NS. Accordingly, we revised this figure and the related discussion in manuscript.

Fig. 4 (a) The geometric structures for pure hcp, polymorphic, and pure fcc RhMo and their formation energies per atom.

[Added results]: [Page 10, Line 225] “The hcp RhMo NS has the highest formation energy (0.9698 eV atom⁻¹) due to its metastable nature; the local edge phase transition from metastable hcp phase to stable fcc phase in the RhMo NS can lower the formation energy (0.9681 eV atom⁻¹) to stabilize the whole configuration and therefore protect the inner metastable phase.”

Fig. 4 (a) The geometric structures for pure hcp, polymorphic, and pure fcc RhMo and their formation energies per atom.

Comment 5. The authors calculated the migration barriers of a hydrogen atom at the hcp/fcc interface using NEB method. It is strange that the site 5 to site 6 step size is much greater than the others. The author should be better to provide in SI the details of the critical structures along the migration path, to ensure that all migration steps are elementary steps along the pathway.

Response: Thank you very much for your useful comment. According to your suggestion, we re-evaluated the hydrogen binding on 10 potential adsorption sites in **Supplementary Fig. 26**. Sites 5 and 6 in **Fig. 4b** point to sites F and H in **Supplementary Fig. 26**. Then 7 local minimum sites were determined as the energy-minimum migration pathway in **Fig. 4a**. Based on our revised data, there are no strange data for H migration at the hcp/fcc interface using NEB method. We also provided the critical NEB structures of H migration, including the initial state, transition state and final state, in **Supplementary Fig. 26b** and **Fig. 4b** for better reproduction. The files about geometry structures along the migration path are also provided via the submission system.

Supplementary Fig. 26 (a) Potential adsorption sites of hydrogen on fcc-hcp polymorph. (b) The corresponding hydrogen adsorption free energies.

Fig. 4 (b) Geometry configurations for seven potential active sites on the surface of RhMo NSs and corresponding hydrogen adsorption free energies on each site.

[Added results]: [Page 10, Line 231] “Accordingly, as shown in **Figure 4b** and **Supplementary Fig. 26**, we evaluated the HBEs of 7 potential active sites on the surface of RhMo NSs and found that the polymorphic interface between the *hcp* and *fcc* phases had a stronger HBE of -0.30 eV, which aided in the dissociation of H_2 to generate $*H$.”

Supplementary Fig. 26 (a) Potential adsorption sites of hydrogen on fcc-hcp polymorph. (b) The corresponding hydrogen adsorption free energies.

Fig. 4 (b) Geometry configurations for seven potential active sites on the surface of RhMo NSs and corresponding hydrogen adsorption free energies on each site.

Comment 6. Too many grammatic errors and typos in the manuscript. Many sentences need to be refined to conform to the logic.

Response: Thank you very much for your careful review. We have already corrected these mistakes to improve the logic. Our paper has also been polished by Nature Research Editing Service to improve the overall quality.

REVIEWERS' COMMENTS

Reviewer #1 (Remarks to the Author):

The authors have carefully addressed all my concerns, therefore it can be accepted now.

Reviewer #3 (Remarks to the Author):

The authors have responded appropriately to the concerns and comments to the previous version of manuscript, with necessary revisions incorporated in the new submission. Considering the quality of present manuscript as well as its scientific importance, the reviewer would recommend it for publication.

Response to Reviewers

Dear Reviewers,

Thanks for your valuable time to constructive comments on our manuscript titled “Atomic-Thick Metastable Phase RhMo Nanosheets for Hydrogen Oxidation Catalysis” for **Nature Communications** (NCOMMS-22-47904-A).

Reviewer #1 (Remarks to the Author): The authors have carefully addressed all my concerns, therefore it can be accepted now.

Response: Thank you for agreeing to publish our manuscript. We also sincerely appreciate your comments and suggestions on our work, which have certainly improved our manuscript.

Reviewer #3 (Remarks to the Author): The authors have responded appropriately to the concerns and comments to the previous version of manuscript, with necessary revisions incorporated in the new submission. Considering the quality of present manuscript as well as its scientific importance, the reviewer would recommend it for publication.

Response: We thank you for the recommendation for the publication of our manuscript.